# Ultrarare Loss-of-Function Mutations in the Genes Encoding the Ionotropic Glutamate Receptors of Kainate Subtypes Associated with Schizophrenia Disrupt the Interaction with PSD95

**DOI:** 10.3390/jpm12050783

**Published:** 2022-05-12

**Authors:** Tsung-Ming Hu, Chia-Liang Wu, Shih-Hsin Hsu, Hsin-Yao Tsai, Fu-Yu Cheng, Min-Chih Cheng

**Affiliations:** 1Department of Psychiatry, Yuli Branch, Taipei Veterans General Hospital, Hualien 98142, Taiwan; tsungming.hu@gmail.com (T.-M.H.); peterwu2000.tw@yahoo.com.tw (C.-L.W.); filvhsu@gmail.com (S.-H.H.); ashleytsai0808@gmail.com (H.-Y.T.); op19910911@gmail.com (F.-Y.C.); 2Department of Future Studies and LOHAS Industry, Fo Guang University, Jiaosi, Yilan County 26247, Taiwan

**Keywords:** kainate receptor, gene, schizophrenia, resequence, rare mutation, PSD95

## Abstract

Schizophrenia is a complex mental disorder with a genetic component. The GRIK gene family encodes ionotropic glutamate receptors of the kainate subtype, which are considered candidate genes for schizophrenia. We screened for rare and pathogenic mutations in the protein-coding sequences of the GRIK gene family in 516 unrelated patients with schizophrenia using the ion semiconductor sequencing method. We identified 44 protein-altered variants, and *in silico* analysis indicated that 36 of these mutations were rare and damaging or pathological based on putative protein function. Notably, we identified four truncating mutations, including two frameshift deletion mutations (*GRIK1*^p.Phe24fs^ and *GRIK1*^p.Thr882fs^) and two nonsense mutations (*GRIK2*^p.Arg300Ter^ and *GRIK4*^p.Gln342Ter^) in four unrelated patients with schizophrenia. They exhibited minor allele frequencies of less than 0.01% and were absent in 1517 healthy controls from Taiwan Biobank. Functional analysis identified these four truncating mutants as loss-of-function (LoF) mutants in HEK-293 cells. We also showed that three mutations (*GRIK1*^p.Phe24fs^, *GRIK1*^p.Thr882fs^, and *GRIK2*^p.Arg300Ter^) weakened the interaction with the PSD95 protein. The results suggest that the GRIK gene family harbors ultrarare LoF mutations in some patients with schizophrenia. The identification of proteins that interact with the kainate receptors will be essential to determine kainate receptor-mediated signaling in the brain.

## 1. Introduction

Schizophrenia is a chronic debilitating mental disorder that affects approximately 1% of the population worldwide, with symptoms appearing in early adulthood or late adolescence. The heritability of schizophrenia is estimated at 80% on average [1]. Evidence from genetic, neuropathological, and meta-analysis studies indicates that a reduction in neuronal processes and synaptic dysfunction is involved in the pathogenesis of schizophrenia, and patients exhibit synaptic degeneration in the brain [2,3,4,5,6]. The pathogenesis of schizophrenia may arise from mutations of synapse-associated genes; thus, identifying such mutations in schizophrenia will provide insight into the cause of schizophrenia and lead to new treatments. Glutamate receptors regulate neurotransmission in the mammalian central nervous system and play an essential role in synaptic plasticity, neurodevelopment, and cognitive function [7]. Glutamate receptor ion channels, including the *N*-methyl-d-aspartic acid (NMDA) receptors, the amino-3-hydroxy-5-methyl-4-isoxazole (AMPA) receptors, and the kainate receptors, are the major excitatory neurotransmitter receptors in the brain. These contribute to various normal neurophysiologic processes [8,9,10].

The kainate receptors, encoded by the GRIK gene family, are composed of four subunits and function as ligand-activated ion channels [11]. Rare pathologic mutations of the GRIK gene family may play a role in conferring susceptibility to schizophrenia for the following reasons: first, the kainate receptors are expressed in the brain and contribute to synaptic plasticity, transmission, learning, memory, and neuropsychiatric disorders, such as autism and schizophrenia [9,12,13,14]; second, several studies have demonstrated abnormal kainate receptor expression in the brains of subjects with schizophrenia [15,16,17,18,19,20,21,22,23]; third, although several studies have demonstrated that no significant associations of genetic markers of the GRIK gene family are associated with schizophrenia [24,25,26,27,28,29,30], some have identified novel rare mutations of the GRIK gene family, suggesting a potential role for rare and significant effects of mutations of the GRIK gene family in the susceptibility to schizophrenia [24,31,32,33,34].

The genetic basis of schizophrenia may involve rare deleterious mutations with high penetrance or multiple common polymorphisms with low penetrance [35,36]. In particular, rare mutations in a group of genes linked to synaptic development, function, and plasticity were identified in patients diagnosed with schizophrenia, suggesting that rare high-risk genetic mutations with result in the dysregulation of synaptic transmission and contribute to the pathophysiology of schizophrenia [37,38,39,40]. Therefore, we sequenced the exonic regions of five kainate receptor genes (*GRIK1*, *GRIK2*, *GRIK3*, *GRIK4*, and *GRIK5*) to identify rare pathogenic variants in patients with schizophrenia using the ion semiconductor sequencing method. We analyzed protein function by immunoblotting, immunocytochemistry, and protein–protein interactions of four protein-truncating mutations using a bioluminescent resonance energy transfer (BRET) assay.

## 2. Materials and Methods

### 2.1. Human Subjects

All subjects were Han Chinese from Taiwan: 516 unrelated patients with schizophrenia (258 males, mean age = 43.0 ± 9.3 years, mean age of onset = 22.0 ± 6.7 years; 258 females, mean age = 48.8 ± 10.2 years, mean age of onset = 22.0 ± 7.6 years) were screened for mutations. Blood samples from patients with schizophrenia was obtained from the Yuli Branch, Taipei Veterans General Hospital. Patients fulfilling the diagnostic criteria for schizophrenia defined by the *Diagnostic and Statistical Manual of Mental Disorders-Fourth Edition Tex Revision* were recruited for the study. The diagnosis of schizophrenia was based on clinical interviews and the review of medical records by senior psychiatrists with consensus. Organic brain syndrome cases, intellectual disability, substance-related psychosis, and mood disorder with psychotic features were excluded. The study was approved by the institutional review board from the Antai Medical Care Cooperation Antai-Tian-Sheng Memorial Hospital (Approval number: 18-144-A). All patients received a full explanation of the study and provided signed written informed consent. Genomic DNA was isolated from peripheral blood cells using the Gentra Genomic DNA Purification kit (QIAGEN, USA) and stored at −20 °C until use.

### 2.2. Ion Semiconductor Sequencing, PCR Reaction, and Fluorescence-Based Cycle Sequencing

Amplification primers were designed to cover the exonic regions of five genes, *GRIK1* (GeneID:2897), *GRIK2* (GeneID:2898), *GRIK3* (GeneID:2899), *GRIK4* (GeneID:2900), and *GRIK5* (GeneID:2901), with the Ion AmpliSeq™ Designer software (Thermo Fisher Scientific Inc., Waltham, MA, USA). Ion semiconductor sequencing, PCR reactions, and fluorescence-based cycle sequencing were performed following previously described methods [41].

### 2.3. In Silico Analysis of Amino Acid Substitutions

The potential functional consequences of amino acid substitution were predicted using the Polyphen-2 (http://genetics.bwh.harvard.edu/pph2/, accessed on 18 July 2019), SIFT (https://sift.bii.a-star.edu.sg/, accessed on 18 July 2019), Pmut (http://mmb.pcb.ub.es/PMut/, accessed on 18 July 2019), PROVEAN (http://provean.jcvi.org/index.php, accessed on 18 July 2019), and CADD (https://cadd.gs.washington.edu/, accessed on 26 April 2022) software tools. Mutations were checked whether they were documented in the NCBI dbSNP database (http://www.ncbi.nlm.nih.gov/SNP/, accessed on 18 July 2019), the Genome Aggregation Database (gnomAD, https://gnomad.broadinstitute.org/, accessed on 18 July 2019), 1000 Genomes Project (https://www.internationalgenome.org/, accessed on 26 April 2022) and the Taiwan BioBank (https://www.twbiobank.org.tw/new_web/index.php, accessed on 18 July 2019). Protein–protein interactions were predicted by the STRING database (version 11.5, https://string-db.org/, accessed on 1 September 2021).

### 2.4. RNA Preparation and Semi-Quantitative RT-PCR

Peripheral blood cells from a non-psychotic control (a lab technician) were used in RNA purification and lymphoblastoid cell line preparation. Lymphoblastoid cell lines were prepared according to a standard protocol established in our laboratory [42]. Total cellular RNA was purified using TRIzol reagent, according to the manufacturer’s protocol (Invitrogen Life Technologies, Carlsbad, CA, USA). Human brain Marathon-Ready cDNA was purchased from Clontech Laboratories, Inc., Mountain View, CA, USA. Complementary DNA was synthesized using SuperScript III™ Reverse Transcriptase (Invitrogen), according to the manufacturer’s instructions. Semi-quantitative RT-PCR was used for the quantitation of GRIK gene family expression. The primer sequences are listed in Appendix A. The PCR products were electrophoresed on 2% agarose gels, stained with ethidium bromide, and photographed. The intensity of the gene band was assessed by NIH ImageJ software (http://rsb.info.nih.gov/nih-image/, accessed on 26 April 2022) and normalized by the intensity of the *GAPDH* gene. Each experiment was repeated three times. The differences in gene expression activity between the two groups were analyzed using Student’s *t*-test. *p* < 0.05 was considered significant for all tests.

### 2.5. Gene Constructs and Transient Transfection of Cell Lines

pCMV6-Myc-DDK-tagged *GRIK1* (RC222898), *GRIK2* (RC222369), and *DLG4* genes (RC215178) were purchased from Origene Technologies, Inc., Rockville, MD, USA. The cDNA encoding *GRIK3*, *GRIK4*, and *GRIK5* were synthesized by MDBio, Inc. (Taiwan) and subcloned into the pCMV6-Myc-DDK vector (Origene). The open reading frames derived from the pCMV6-Myc-DDK vector were cloned into pCMV6-AC-GFP (Origene) with a C-terminal tGFP tag using the *SgfI* and *MluI* enzymes (New England Biolabs, Ipswich, MA, USA). The *DLG4* cDNA was cloned into pFN31K-Nluc-CMV-neo Flexi^®^, whereas the *GRIK1*, *GRIK2*, and *GRIK4* cDNAs were cloned into pFN21A-HaloTag-CMV-Flexi by Flexi^®^ Vector Systems (Promega, Madison, WI, USA), according to the manufacturer’s protocol. The mutant-type constructs were generated using the QuikChange^®^ Lightning Site-Directed Mutagenesis kit (Agilent Technologies, Santa Clara, CA, USA), according to the manufacturer’s instructions. The authenticity of the cloned sequences was verified by qualitative restriction enzyme digestion and Sanger sequencing.

Human embryonic kidney cells (HEK-293 cells, ATCC^®^ CRL1573™) were purchased from the Bioresource Collection and Research Center, Food Industry Research and Development Institute, Taiwan, and cultured in the complete minimum essential medium supplemented with 10% FBS and 1% antibiotics at 37 °C in a 5% CO_2_ environment. The cells were cultured in 6-well plates for immunoblot analysis and transfected with constructs using Lipofectamine™ 3000 (Invitrogen, Waltham, MA, USA).

### 2.6. Immunoblot Assay

HEK-293 cells were homogenized in 500 μL lysis solution containing 100 mM Tris-HCl (pH = 9.0), 100 mM NaCl, 0.5% (*v*/*v*) Triton X-100, and a protease inhibitor cocktail (Roche Diagnostics GmbH, Mannheim, Germany). The homogenates were centrifuged at 13,000 rpm for 60 min at 4 °C, and the supernatants were stored at −80 °C until use. Protein concentrations were determined by the Coomassie protein assay (Pierce Biotechnology Inc., Rockford, IL, USA). Immunoblotting analysis was performed according to standard protocols with the following primary antibodies: rabbit anti-GRIK1 (ARG55272, arigo Biolaboratories Corp, Taiwan), rabbit anti-GRIK2 (tcua3837, Taiclone, Taiwan), rabbit anti-GRIK3 (A10701, ABclonal, Woburn, MA, USA), rabbit anti-GRIK4 (tcba3836, Taiclone), mouse anti-tGFP (TA150041, Origene), and mouse anti-GAPDH (G8795, Sigma-Aldrich, Saint Louis, MO, USA). Anti-mouse IgG and goat anti-rabbit IgG conjugated with horseradish peroxidase were used as the secondary antibodies. The chemiluminescence signal was visualized using an ECL detection system (GE Healthcare Bio-Sciences AB, Uppsala, Sweden). The intensity of the immunoblot was assessed with NIH ImageJ software (http://rsb.info.nih.gov/nih-image/, accessed on 1 April 2019).

### 2.7. Immunocytochemistry

Cultured cells were fixed in 4% paraformaldehyde in PBS (pH 7.4) for 20 min at room temperature, washed three times with PBS containing 0.1% Triton X-100, and blocked for 40 min in PBS containing 1% bovine serum albumin and 0.1% Triton X-100. The samples were stained with Alexa Fluor^®^ (Invitrogen, Illkirch-Graffenstaden, France) 594 wheat germ agglutinin (WGA) (Invitrogen) and 4′,6-diamidino-2-phenylindole (DAPI). Images were acquired with an Axio Vert.A1 (Zeiss, Jana, Germany) fluorescence microscope and processed with ZEN 2 software (Zeiss).

### 2.8. BRET Assay

The NanoBRET™ assay was performed in white 96-well plates, according to the manufacturers’ protocols (Promega). Briefly, both NanoLuc^®^ and HaloTag^®^ fusion vectors were transfected into HEK-293 cells. After a 24 h incubation, the cells were seeded into white 96-well plates with and without NanoBRET™ HaloTag^®^ 618 Ligand. Following a 24 h incubation, NanoBRET NanoGlo substrate (Promega) was added (0.1 μL/well) and readings were performed for 0.3 s using NanoLuc^®^ emission (460 nm) to measure the donor signal and NanoBRET™ ligand emission (620 nm) for the acceptor signal using a Varioskan Flash instrument (Thermo Scientific). BRET was calculated as the ratio of the emission at 620 nm/460 nm. Each experiment was performed at least six times.

## 3. Results

### 3.1. Identification of the Rare and Protein-Altering Variants of the GRIK Gene Family in Patients with Schizophrenia

We designed 158 amplicons (97.4% overall coverage) to cover the exons of *GRIK1*, *GRIK2*, *GRIK3*, *GRIK4*, and *GRIK5*. After resequencing 516 unrelated patients with schizophrenia, 44 protein-altering variants were identified, including 12 in the *GRIK1*, 5 in the *GRIK2*, 7 in the *GRIK3*, 13 in the *GRIK4*, and 7 in the *GRIK5* (Table 1, Appendix A). Of these, 36 missense mutations had minor allele frequencies (MAFs) less than 0.5% in the gnomAD database and Taiwan BioBank. Five variants (*GRIK1*^p.Phe24fs^, *GRIK1*^p.Arg203Leu^, *GRIK1*^p.His336Arg^, *GRIK4*^p.Gln342Ter^, and *GRIK4*^p.His403Gln^) were not observed among 1517 healthy controls from the Taiwan BioBank, 1000 Genomes Project, and the gnomAD database. *In silico* analysis revealed that the five mutations (*GRIK1*^p.Arg203Leu^, *GRIK1*^p.Arg558Gln^, *GRIK2*^p.His253Tyr^, *GRIK4*^p.Arg507Trp^, and *GRIK4*^p.Arg760Pro^) were predicted to be pathogenic using the Polyphen-2, SIFT, Pmut, and PROVEAN prediction programs (Table 1, bold text). We identified two heterozygous frameshift deletion mutations (*GRIK1*^p.Phe24fs^ and *GRIK1*^p.Thr882fs^) in two unrelated patients with schizophrenia and two heterozygous nonsense mutations (*GRIK2*^p.Arg300Ter^ and *GRIK4*^p.Gln342Ter^) in two unrelated patients with schizophrenia. The two mutants (*GRIK1*^p.Phe24fs^ and *GRIK1*
^p.Thr882fs^) were predicted to shift the reading frames, resulting in a completely different protein translation from the original GRIK1 protein (Figure 1A), whereas the two mutants (*GRIK2*^p.Arg300Ter^ and *GRIK4*^p.Gln342Ter^) were predicted to introduce a stop codon in the coding region (Figure 1B). Three mutations (*GRIK1*^p.Phe24fs^, *GRIK2*^p.Arg300Ter^, and *GRIK4*^p.Gln342Ter^) resulted in a truncated protein lacking a predicted gene transmembrane domain of the kainate receptor (Figure 1C).

### 3.2. GRIK Gene Family mRNA Expressions in Peripheral Blood Cells, Lymphoblastoid Cells, and Human Brain

We measured the expression of the *GRIK1*, *GRIK2*, *GRIK3*, *GRIK4*, and *GRIK5* genes in peripheral blood cells and lymphoblastoid cells of a non-psychotic control and human brain specimens using semi-quantitative RT-PCR. The *GRIK1*, *GRIK2*, *GRIK3*, *GRIK4*, and *GRIK5* genes were detected in human brain tissue, but not significantly expressed in peripheral blood cells and lymphoblastoid cells compared with human brain tissue (Appendix A).

### 3.3. Immunoblotting and Immunocytochemistry Analysis of GRIK Gene Family Mutants in Cultured Cells

To understand the effects of protein-altering mutations on GRIK protein expression and translocation, we examined the expression and subcellular localization of the wild-type and mutant GRIK proteins in HEK-293 cells. We cloned the wild-type cDNAs of the GRIK gene family into a mammalian vector with a C-terminal tGFP tag, pCMV6-AC-GFP. Immunoblotting was used to confirm GRIK gene family protein overexpression in lysates using anti-tGFP (Figure 2A). We determined the effects of GRIK gene family mutants on protein expression using an immunoblot assay. HEK-293 cells were transfected with wild-type (WT) or mutant constructs. After 24 h of transfection, the four mutant-type constructs (*GRIK1*^p.Phe24fs^, *GRIK1*^p.Thr882fs^, *GRIK2*^p.Arg300Ter^, and *GRIK4*^p.Gln342Ter^) did not express GRIK-tGFP fusion proteins in HEK-293 cells compared with the WT constructs (Figure 2B,C,E). Immunoblotting of the cell lysates from the HEK-293 cells carrying *GRIK3*^p.Arg215His^ and *GRIK3*^p.Thr867Cys^ revealed significantly decreased GRIK3-tGFP fusion proteins in the mutants relative to the WT cells (Figure 2D). HEK-293 cells carrying *GRIK5* mutants demonstrated similar levels of GRIK5-tGFP fusion protein expression as the WT cells (Figure 2F). Immunocytochemical analysis revealed the four mutant-type constructs (*GRIK1*^p.Phe24fs^, *GRIK1*^p.Thr882fs^, *GRIK2*^p.Arg300Ter^, and *GRIK4*^p.Gln342Ter^) did not express the recombinant protein in HEK-293 cells (Appendix A).

### 3.4. Protein-Truncating Mutations in the GRIK Gene Family Disrupt the Interaction with the Post-Synaptic Density Protein 95 (PSD95) Protein

The STRING database was used to identify interactions of the GRIK gene family and to map the protein–protein interaction network with an integrated confidence score of 0.400. The STRING results revealed that all GRIK gene members (red frame) interact with the *DLG4* gene encoding the PSD95 protein (green frame, Figure 3). The PSD95 protein was used for in vitro protein–protein interaction analysis based on the STRING results and a literature review.

NanoBRET assay used a NanoLuc^®^ fusion protein as the energy donor and a fluorescently labeled HaloTag™ fusion protein as the energy acceptor. First, we used the NanoBRET assay to detect GRIK gene family/PSD95 interactions in live cells. We found that GRIK1, GRIK2, and GRIK4 proteins were associated with PSD95 by measuring the energy transfer from a bioluminescent protein donor to a fluorescent protein acceptor (Figure 4A). Next, we evaluated the impact of four ultrarare truncated GRIK mutations (*GRIK1*^p.Phe24fs^, *GRIK1*^p.Thr882fs^, *GRIK2*^p.Arg300Ter^, and *GRIK4*^p.Gln342Ter^) on the interaction among themselves and with the PSD95 protein. We observed that the interactions of the three truncated GRIK mutant proteins (*GRIK1*^p.Phe24fs^, *GRIK1*^p.Thr882fs^, and *GRIK2*^p.Arg300Ter^) were significantly weakened or even abolished with PSD95, whereas the interaction of wild-type GRIK protein with PSD95 protein was preserved (Figure 4B–D).

## 4. Discussion

We identified 4 truncating mutants and 36 rare missense mutations with MAFs less than 0.5% in the GRIK gene family from 516 patients with schizophrenia, including 5 variants (*GRIK1*^p.Phe24fs^, *GRIK1*^p.Arg203Leu^, *GRIK1*^p.His336Arg^, *GRIK4*^p.Gln342Ter^, and *GRIK4*^p.His403Gln^) that were not observed among 1517 healthy controls from the Taiwan BioBank, the gnomAD database, and 1000 Genomes Project. We presumed that these are ultrarare variants, which may contribute to the pathogenesis of schizophrenia. To interpret the clinical relevance of the protein-altering mutations, we planned to directly evaluate the expression changes in the GRIK gene family in peripheral blood cells and lymphoblastoid cells of patients. We measured the expression of the GRIK gene family members in peripheral blood cells and lymphoblastoid cells from a non-psychotic control; however, they were not significantly expressed, suggesting that the GRIK gene family is not expressed in these cell types. Thus, we did not perform expression assays in peripheral blood cells and lymphoblastoid cells from patients with schizophrenia. We confirmed the four loss-of-function (LoF) mutants (*GRIK1*^p.Phe24fs^, *GRIK1*^p.Thr882fs^, *GRIK2*^p.Arg300Ter^, and *GRIK4*^p.Gln342Ter^) in HEK-293 cells as they did not express GRIK-tGFP fusion proteins compared with the wild-type constructs, and two missense mutants (*GRIK3*^p.Arg215His^ and *GRIK3*^p.Thr867Cys^) were significantly decreased in expression in HEK-293 cells. Notably, the four protein-truncating mutations (*GRIK1*^p.Phe24fs^, *GRIK1*^p.Thr882fs^, *GRIK2*^p.Arg300Ter^, and *GRIK4*^p.Gln342Ter^) and two missense mutations (*GRIK3*^p.Arg215His^ and *GRIK3*^p.Thr867Cys^) identified in this study represent ultrarare variants that have not been reported in the literature, to the best of our knowledge. We presumed that these ultrarare mutations impair protein function and alter biological processes of synaptic function in patients with schizophrenia. However, *in silico* analysis of these two missense mutations (*GRIK3*^p.Arg215His^ and *GRIK3*^p.Thr867Cys^) denied this possibility. Thus, the effects of these two missense mutations on the pathophysiology of schizophrenia still need to be fully elucidated. Given the premature termination of protein synthesis or reduction in the transcript levels by nonsense and frameshift-mediated decay [43,44], the three predicted mutations (*GRIK1*^p.Phe24fs^, *GRIK2*^p.Arg300Ter^, and *GRIK4*^p.Gln342Ter^) lack the adjacent transmembrane domain, which results in a complete loss of function of the kainate receptor. Taken together, we presumed that the rare protein-altering mutations, especially LoF mutations, in genes encoding kainate receptors are associated with the molecular pathology of schizophrenia.

Several synapse-associated proteins, especially PSD95 protein, were identified as interacting proteins for the kainate receptors [45,46,47]. For example, co-immunoprecipitations using rat brain extracts showed that GRIK2 and GRIK5 proteins were associated with the PSD95 protein [46]. In addition, the results from the STRING database indicated that all GRIK gene family members interact with the PSD95 protein. PSD95 is involved in the maturation of synapse formation and genetic studies have indicated that genetic variants of PSD95 are associated with increased susceptibility to schizophrenia [48,49]. Therefore, we speculated that the four LoF mutations (*GRIK1*^p.Phe24fs^, *GRIK1*^p.Thr882fs^, *GRIK2*^p.Arg300Ter^, and *GRIK4*^p.Gln342Ter^) that we identified from patients with schizophrenia may affect the binding of the kainate receptors with PSD95 protein. Based on the results from our NanoBRET assay, GRIK1, GRIK2, and GRIK4 bind to PSD95. Moreover, we experimentally confirmed that two frameshift deletion mutations (*GRIK1*^p.Phe24fs^ and *GRIK1*^p.Thr882fs^) and one nonsense mutation (*GRIK2*^p.Arg300Ter^) identified in patients with schizophrenia weaken or even disrupt the interaction between GRIK and PSD95 proteins. Thus, we speculated that three out of these four ultrarare LoF mutations (*GRIK1*^p.Phe24fs^, *GRIK1*^p.Thr882fs^, and *GRIK2*^p.Arg300Ter^) may affect the interaction of PSD95 protein with kainate receptors in the brain and consequently affect synaptic function in patients with schizophrenia. Further analysis of these mutations is needed to confirm our hypothesis.

Several studies have demonstrated shared predisposing genetic factors and common mechanisms in neurodevelopmental pathways between different psychiatric disorders [50,51,52]. Cordoba et al. identified a nonsense mutation (*GRIK2*^p.Arg198*^) in the *GRIK2* gene as a basis for the etiology of intellectual disabilities [53]. In addition, recurrent de novo variants (*GRIK2*^p.Ala657Thr^, *GRIK2*^p.Thr660Lys^, and *GRIK2*^p.Thr660Arg^) in the *GRIK2* gene were identified in various neurodevelopmental disorders [54,55]. In addition, several studies found that the *GRIK2* gene may be associated with autism spectrum disorder [56,57,58]. For example, Jamain et al. identified a missense mutation, *GRIK2*^p.Met867Ile^, in a highly conserved domain of the intracytoplasmic C-terminal region of the GRIK2 protein in patients with autism [58]. A case report of a girl with severe developmental delay revealed a 2.6 Mb microdeletion in 1p34.3 involving the *GRIK3* gene [59]. Recently, another study found that damaging coding variants within the kainate receptor ion channel genes confers increased risk for schizophrenia, autism, and intellectual disabilities [60]. In this study, we identified several ultrarare missense variants in the GRIK gene family from patients with schizophrenia and demonstrated altered protein expression and weakened interactions with the PSD95 protein. These studies and our results support the existence of shared biological pathways in schizophrenia, autism, and intellectual disability.

In summary, the GRIK gene family may harbor ultrarare variants, particularly those deleterious to protein expression in certain patients with schizophrenia, supporting the idea that rare coding variants may contribute to the genetic architecture of schizophrenia. Furthermore, identifying proteins that interact with kainate receptors is essential to unravel kainate-receptor-mediated signaling in the brain. Future proteomic studies using cell models carrying these rare deleterious mutations are necessary to understand the role of kainate-receptor-interacting proteins and how they contribute to the etiology of schizophrenia.

## Figures and Tables

**Figure 1 jpm-12-00783-f001:**
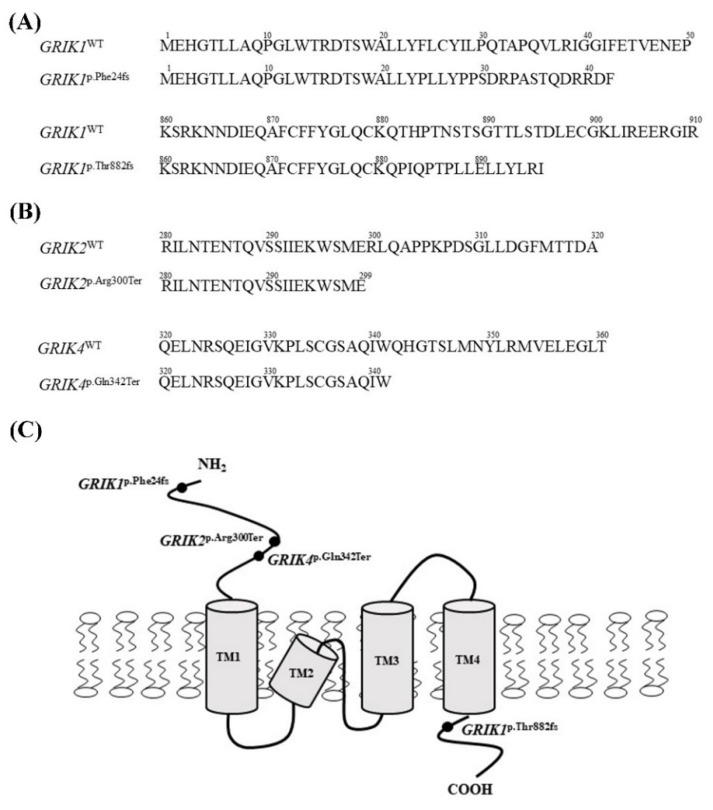
Four schizophrenia-associated and rare protein-truncating mutations identified in this study. (**A**) Schematic representation of the two frameshift mutants (*GRIK1*^p.Phe24fs^ and *GRIK1*^p.Thr882fs^) of the *GRIK1*. (**B**) Schematic representation of the two nonsense mutants (*GRIK2*^p.Arg300Ter^ and *GRIK4*^p.Gln342Ter^) in *GRIK2* and the *GRIK4*, respectively. (**C**) Schematic representation of the kainate receptor structure and locations of the four rare protein-truncating mutations of the GRIK gene family. TM, transmembrane.

**Figure 2 jpm-12-00783-f002:**
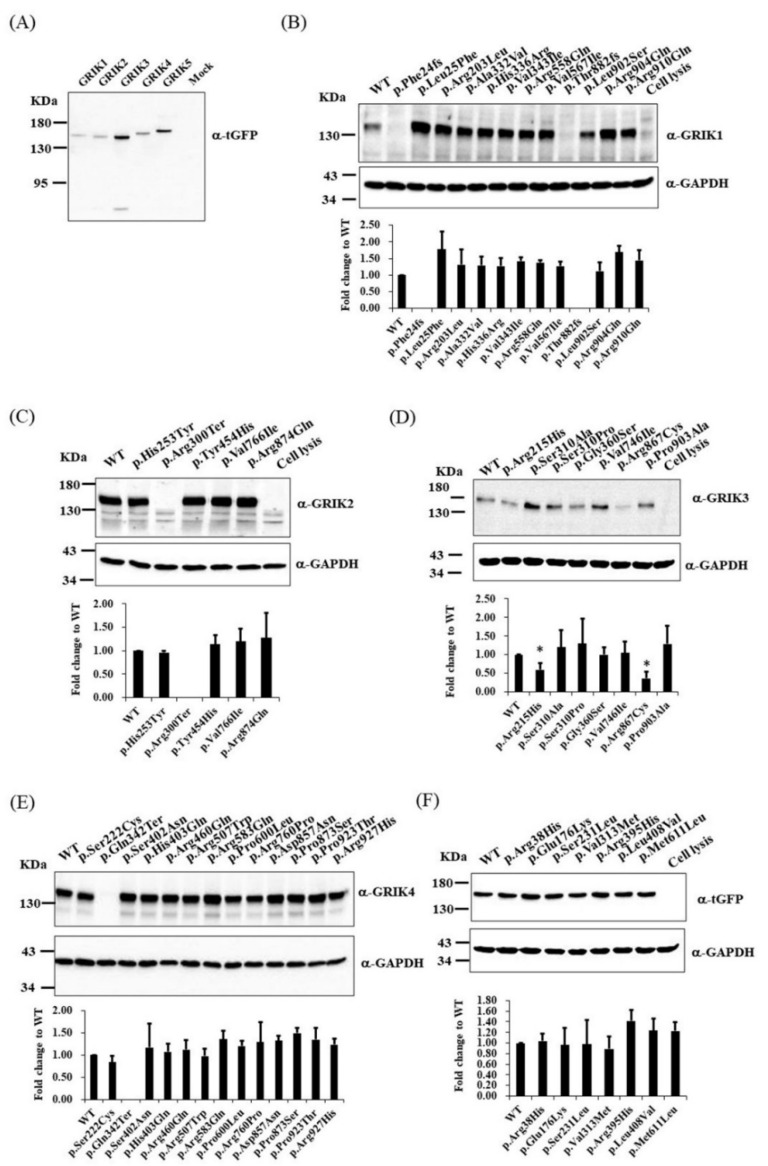
Immunoblotting of lysates extracted from HEK-293 transiently co-transfected with plasmids expressing either WT GRIK cDNA or mutant cDNAs. (**A**) Immunoblotting validated GRIK gene family protein overexpression in lysates using anti-tGFP. (**B**) *GRIK1* WT versus 12 mutants. (**C**) *GRIK2* WT versus five mutants. (**D**) *GRIK3* WT versus seven mutants. (**E**) *GRIK4* WT versus 13 mutants. (**F**) *GRIK5* WT versus seven mutants. For normalization, the lysates were analyzed in parallel by anti-GAPDH immunoblotting. The results are representative of three independent experiments. WT, wild-type. Graphs represent means ± standard deviation. * *p* < 0.05.

**Figure 3 jpm-12-00783-f003:**
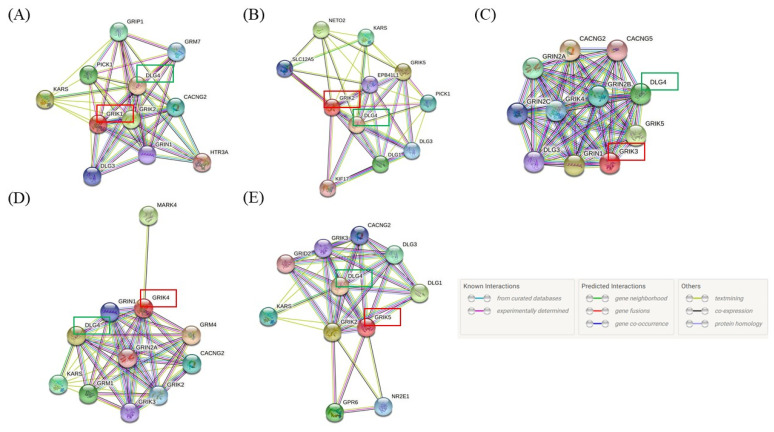
Graphical representation of the GRIK gene family network. (**A**) *GRIK1*: number of nodes: 11; PPI enrichment *p*-value: 5.66 × 10^−14^. (**B**) *GRIK2*: number of nodes: 11; PPI enrichment *p*-value: 7.15 × 10^−11^. (**C**) *GRIK3*: number of nodes: 11; PPI enrichment *p*-value: <1.0 × 10^−16^. (**D**) *GRIK4*: number of nodes: 11; PPI enrichment *p*-value: 3.79 × 10^−12^. (**E**) *GRIK5*: number of nodes: 11; PPI enrichment *p*-value: 8.92 × 10^−10^.

**Figure 4 jpm-12-00783-f004:**
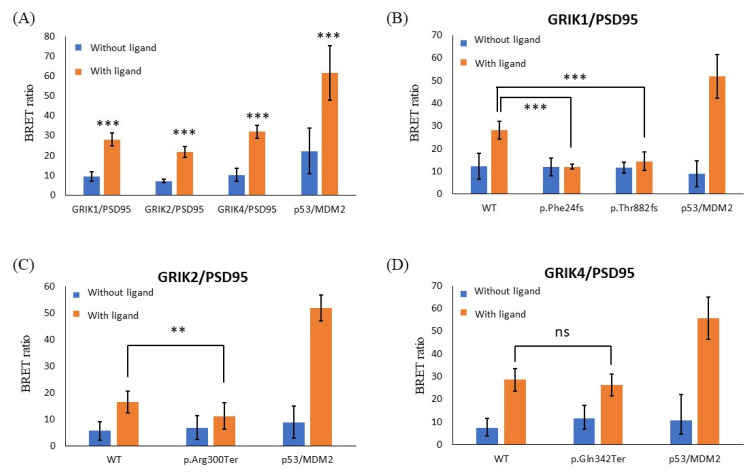
NanoBRET assay to detect the interaction between GRIK and PSD95 protein in live cells. (**A**) GRIK1, GRIK2, and GRIK4 proteins were associated with PSD95 protein by measuring energy transfer from a bioluminescent protein donor to a fluorescent protein acceptor. The p53/MDM2 was a positive control. Differences in BRET ratio between with and without NanoBRET™ HaloTag^®^ 618 Ligands were assessed using Student’s *t*-tests. (**B**,**C**) The three mutants (*GRIK1*^p.Phe24fs^, *GRIK1*^p.Thr882fs^, and *GRIK2*^p.Arg300Ter^) significantly weakened or even abolished the interaction among themselves and with the PSD95 protein, whereas the interaction of GRIK WT proteins with PSD95 was preserved. (**D**) The interactions of GRIK4 WT and mutant-type (*GRIK4*^p.Gln342Ter^) proteins with PSD95 were preserved. Graphs represent means ± standard deviation. *** *p* < 0.001, ** *p* < 0.01, ns means nonsignificant.

**Table 1 jpm-12-00783-t001:** Bioinformatic analysis of the protein-altering mutations identified from 516 patients with schizophrenia in the semiconductor sequencing stage.

Genomic Position	Variant (Amino Acid Change)	Patient Number	dbSNP ID	MAF			In Silico Analysis for Amino Acid Substitution				
				gnomAD (Non-Neuro)	1000G	Taiwan Biobank	Polyphen-2	SIFT	Pmut	PROVEAN	CADD Raw/PHRED
*GRIK1*											
chr21:31311747	c.70_71delTT (p.Phe24fs)	1	N/R	N/R	N/R	N/R	N/A	N/A	N/A	N/A	N/A
chr21:31311746	c.73C > T (p.Leu25Phe)	2	rs200262014	0.0002	0.0008	0.0011	Benign	Tolerated	Neutral	Neutral	2.15/21.1
**chr21:31045421**	**c.608G > T (p.Arg203Leu)**	**1**	**N/R**	**N/R**	**N/R**	**N/R**	**Probably damaging**	**Damaging**	**Disease**	**Deleterious**	**4.27/30**
chr21:31015249	c.995C > T (p.Ala332Val)	1	rs143252117	0.0001	0.0004	0.0007	Benign	Damaging	Disease	Deleterious	3.56/25.2
chr21:31015237	c.1007A > G (p.His336Arg)	1	rs756781607	N/R	N/R	N/R	Benign	Tolerated	Neutral	Neutral	2.51/22.6
chr21:31015217	c.1027G > A (p.Val343Ile)	1	rs1459657790	<0.0001	N/R	N/R	Benign	Tolerated	Neutral	Neutral	2.33/22.1
**chr21:30959806**	**c.1673G > A (p.Arg558Gln)**	**1**	**rs776162409**	**<0.0001**	**N/R**	**N/R**	**Probably damaging**	**Damaging**	**Disease**	**Deleterious**	**3.90/26.9**
chr21:30959780	c.1699G > A (p.Val567Ile)	1	rs536890189	0.0001	0.0006	N/R	Probably damaging	Damaging	Disease	Neutral	3.63/25.5
chr21:30925988	c.2644delA (p.Thr882fs)	1	rs867634999	<0.0001	N/R	N/R	N/A	N/A	N/A	N/A	N/A
chr21:30925928	c.2705T > C (p.Leu902Ser)	41	rs363504	0.0694	0.1098	0.0639	Benign	Tolerated	Neutral	Neutral	2.00/19.76
chr21:30925922	c.2711G > A (p.Arg904Gln)	1	rs769191901	<0.0001	N/R	N/R	Benign	Damaging	Neutral	Neutral	2.62/22.8
chr21:30925904	c.2729G > A (p.Arg910Gln)	3	rs372106416	0.0002	0.0002	0.0023	Benign	Tolerated	Neutral	Neutral	2.55/22.6
*GRIK2*											
**chr6:102130461**	**c.757C > T (p.His253Tyr)**	**4**	**rs770586258**	**<0.0001**	**N/R**	**0.0007**	**Probably damaging**	**Damaging**	**Disease**	**Deleterious**	**4.09/28.5**
chr6:102134175	c.898C > T (p.Arg300Ter)	1	rs1351417917	<0.0001	N/R	N/R	N/A	N/A	N/A	N/A	N/A
chr6:102307204	c.1360T > C (p.Tyr454His)	8	rs186727716	<0.0001	0.0006	0.0020	Benign	Tolerated	Neutral	Neutral	2.29/22
chr6:102483426	c.2296G > A (p.Val766Ile)	7	rs3213608	0.0034	0.00639	0.0026	Benign	Tolerated	Neutral	Neutral	2.23/21.6
chr6:102516280	c.2621G > A (p.Arg874Gln)	2	rs267600750	<0.0001	N/R	0.0003	Possibly damaging	Damaging	Neutral	Neutral	3.23/24.1
*GRIK3*											
chr1:37337877	c.644G > A (p.Arg215His)	1	rs755366301	<0.0001	N/R	N/R	Probably damaging	Damaging	Neutral	Deleterious	3.90/26.9
chr1:37325477	c.928T > G (p.Ser310Ala)	49	rs6691840	0.2747	0.30611	0.0455	Benign	Tolerated	Neutral	Neutral	0.54/9.832
chr1:37325477	c.928T > C (p.Ser310Pro)	2	rs6691840	0.0001	N/R	0.0017	Benign	Tolerated	Neutral	Neutral	0.60/10.34
chr1:37324735	c.1078G > A (p.Gly360Ser)	2	rs551525926	0.0002	0.0004	0.0049	Benign	Tolerated	Neutral	Neutral	2.79/23.1
chr1:37271783	c.2236G > A (p.Val746Ile)	1	rs142411639	<0.0001	N/R	0.0003	Benign	Tolerated	Neutral	Neutral	1.41/15.77
chr1:37267613	c.2599C > T (p.Arg867Cys)	2	rs758481140	<0.0001	N/R	0.0003	Probably damaging	Tolerated	Neutral	Deleterious	3.26/24.2
chr1:37267505	c.2707C > G (p.Pro903Ala)	1	rs777066849	<0.0001	N/R	0.0010	Benign	Damaging	Neutral	Neutral	2.96/23.5
*GRIK4*											
chr11:120702714	c.665C > G (p.Ser222Cys)	1	rs747558889	<0.0001	N/R	0.0004	Benign	Damaging	Neutral	Deleterious	3.06/23.7
chr11:120744892	c.1024C > T (p.Gln342Ter)	1	N/R	N/R	N/R	N/R	N/A	N/A	N/A	N/A	N/A
chr11:120769281	c.1205G > A (p.Ser402Asn)	1	rs776598317	<0.0001	N/R	N/R	Benign	Tolerated	Neutral	Neutral	1.82/18.37
chr11:120769285	c.1209C > A (p.His403Gln)	1	N/R	N/R	N/R	N/R	Benign	Tolerated	Neutral	Neutral	0.89/12.66
chr11:120776105	c.1379G > A (p.Arg460Gln)	1	rs777568733	<0.0001	N/R	0.0003	Benign	Damaging	Neutral	Neutral	2.16/21.1
**chr11:120811098**	**c.1519C > T (p.Arg507Trp)**	**1**	**rs747648319**	**<0.0001**	**N/R**	**0.0003**	**Probably damaging**	**Damaging**	**Disease**	**Deleterious**	**4.41/29.0**
chr11:120827536	c.1748G > A (p.Arg583Gln)	1	rs367593579	<0.0001	N/R	0.0003	Probably damaging	Tolerated	Neutral	Neutral	3.79/26.2
chr11:120827587	c.1799C > T (p.Pro600Leu)	1	rs766921998	<0.0001	N/R	N/R	Probably damaging	Tolerated	Disease	Deleterious	4.16/29.2
**chr11:120837916**	**c.2279G > C (p.Arg760Pro)**	**1**	**rs199658262**	**<0.0001**	**0.0002**	**N/R**	**Probably damaging**	**Damaging**	**Disease**	**Deleterious**	**4.42/32**
chr11:120856667	c.2569G > A (p.Asp857Asn)	8	rs536558955	0.0004	0.0008	0.0100	Benign	Tolerated	Neutral	Neutral	2.87/23.3
chr11:120856715	c.2617C > T (p.Pro873Ser)	3	rs1485463780	<0.0001	N/R	0.0005	Benign	Tolerated	Neutral	Neutral	1.92/19.14
chr11:120856865	c.2767C > A (p.Pro923Thr)	1	rs772831685	N/R	N/R	0.0005	Possibly damaging	Tolerated	Neutral	Neutral	2.87/23.3
chr11:120856878	c.2780G > A (p.Arg927His)	2	rs537835045	<0.0001	0.0002	0.0015	Probably damaging	Damaging	Neutral	Neutral	4.45/32
*GRIK5*											
chr19:42569506	c.113G > A (p.Arg38His)	3	rs143068269	0.0004	0.0004	N/R	Probably damaging	Damaging	Neutral	Neutral	4.13/28.9
chr19:42563662	c.526G > A (p.Glu176Lys)	3	rs201724297	0.0002	0.0010	0.0030	Benign	Tolerated	Neutral	Neutral	3.04/23.7
chr19:42561126	c.692C > T (p.Ser231Leu)	1	rs557315821	<0.0001	0.0002	0.0023	Probably damaging	Tolerated	Neutral	Neutral	2.83/23.2
chr19:42558591	c.937G > A (p.Val313Met)	8	rs746539777	<0.0001	N/R	0.0023	Probably damaging	Tolerated	Neutral	Neutral	3.18/24.0
chr19:42557839	c.1184G > A (p.Arg395His)	2	rs138759162	0.0003	0.0006	0.0010	Possibly damaging	Tolerated	Neutral	Neutral	2.97/23.5
chr19:42557801	c.1222C > G (p.Leu408Val)	1	rs749428580	<0.0001	N/R	N/R	Benign	Tolerated	Neutral	Neutral	1.33/15.37
chr19:42525493	c.1831A > T (p.Met611Leu)	1	rs772424091	<0.0001	N/R	0.0010	Benign	Tolerated	Neutral	Neutral	2.27/21.9

Reference version: GRCh37; *GRIK1*: NM_000830.4; *GRIK2*: NM_021956.4; *GRIK3*: NM_000831.3; *GRIK4*: NM_014619.4; *GRIK5*: NM_002088.4; N/R: not registered; N/A: not available; Polyphen-2: http://genetics.bwh.harvard.edu/pph2/, accessed on 18 July 2019; SIFT: https://sift.bii.a-star.edu.sg/, accessed on 18 July 2019; Pmut: http://mmb.irbbarcelona.org/PMut/, accessed on 18 July 2019; PROVEAN: http://provean.jcvi.org/index.php, accessed on 18 July 2019; CADD: https://cadd.gs.washington.edu/, accessed on 26 April 2022.

## Data Availability

The raw data are available upon request of the corresponding author.

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
