# Peer review of "Ultrarare Loss-of-Function Mutations in the Genes Encoding the Ionotropic Glutamate Receptors of Kainate Subtypes Associated with Schizophrenia Disrupt the Interaction with PSD95"

_jpm, 2022, doi:10.3390/jpm12050783_

Round 1
Reviewer 1 Report
The authors identified rare and pathogenic mutations of the GRIK gene family in patients with schizophrenia. They characterize these mutations and their putative protein products. The manuscript is well organized but there are some issues to review:
- All genes are written correctly in italics except for GRIK followed by gene family in several parts of the work. I suggest putting it in italics.
- Line 80: you write “…written informed consent forms”. I suggest removing forms.
- Materials and methods: lymphoblastoid cells are not mentioned in M&M but in Results section they are used for RNA expression (section 3.2). Please add these samples in M&M and mention where they come from. Are they cell lines or did you obtain them from subjects and further immortalized them in vitro?
- In M&M at 2.4 section I suggest adding the cells you used for semi-quantitative RT-PCR, so it is easier to follow the assays organization.
- At 2.6 section of M&M: I suggest adding at the beginning that immunoblot assays were performed using HEK-293 cells in order to be clear to the reader that these assays were done with this cell line.
- In the results you say that four mutations (frameshift and nonsense) were not expressed in HEK-293 cell line whereas two missense mutations are low expressed. Are these two missense mutations already known in the literature? As you mention these mutants are not considered pathogenic using the predictive programs, but their decreased expression may indicate their involvement in pathology or highlight their tolerance since they are also detected in healthy controls. Can you briefly discuss this aspect in the discussion when you mentioned these mutants?
- In Table 1 I suggest highlighting the five pathogenic mutations.
- Results (Section 3.2): How do you calculate the statistical differences in semi-quantitative RT-PCR? Which are the replicates? Please add this information in M&M.
- As said in the discussion the expression assay has been performed using peripheral blood cells and lymphobastoid cells from subjects. Do you mean from the 516 patients? Are lymphoblastoid cells obtained in vitro through infection with Epstein–Barr virus or is it a cell line you purchased? It is not clear which sample type you use for expression assays as I mentioned before. Please, add it in M&M.
- Section 3.3 (Line 211-212) and discussion (Line 272-274): what do you intend by normal-sized protein products? Is it referred to the molecular weight? Or do you mean expression level since they are low expressed? I suggest changing the word normal-sized because it may be misleading.
- Section 3.3 The figure reference is wrong. Should be Figure 2, not 3 (line 211, 212 and 215)
- Figure 4A: the statistical significance is between the positive orange column and the negative blue column of each interaction, right? Please, specify.
- In line 296-298 you write: “Moreover, we confirmed experimentally that the nonsense mutations in the GRIK genes identified in patients with schizophrenia weaken or even disrupt the interaction between GRIK and PSD95 proteins”. The NanoBRET assay you performed shows that GRIK1p.Phe24fs, GRIK1p.Thr882fs, GRIK2p.Arg300Ter has a weak or even abolished interaction with PSD95. These GRIK1 variants present frameshift mutations while GRIK2 is a nonsense mutation. Therefore, in the aforementioned statement (line 296-298) you are excluding frameshift mutations. So, you should add frameshift and nonsense mutations or leave just mutations.
- In line 299-300 you say: “Thus, we speculated that these four ultrarare LoF mutations may affect the interaction of PSD95 protein with the kainate receptors in the brain…” Why do you include the mutant-type GRIK4p.Gln342Ter if its interaction with PSD95 is preserved? You may say three out of four.
Reviewer 2 Report
My suggestions:
- In the part "Human subjects" I would add a table in the involved individuals (number of male/female individuals, age/age of onset of patients, number of patients with particular mental dysfunctions).
- I would also add for in silico analysis the CADD scores for the variants
- Were the variants checked in 1000Genomes?
- Were there any mutations described in DLG4 gene?
- Is it a chance that GRIK genes may impact other diseases, including neurodegenerative diseases?
Round 2
Reviewer 2 Report
The authors fulfilled my suggestions, thank you.